# Decreased Serum Stromal Cell-Derived Factor-1 in Patients with Familial Hypercholesterolemia and Its Strong Correlation with Lipoprotein Subfractions

**DOI:** 10.3390/ijms242015308

**Published:** 2023-10-18

**Authors:** Lilla Juhász, Hajnalka Lőrincz, Anita Szentpéteri, Nóra Tóth, Éva Varga, György Paragh, Mariann Harangi

**Affiliations:** 1Division of Metabolic Diseases, Department of Internal Medicine, Faculty of Medicine, University of Debrecen, 4032 Debrecen, Hungary; 2Doctoral School of Health Sciences, Faculty of Public Health, University of Debrecen, 4032 Debrecen, Hungary; 3Department of Internal Medicine and Hematology, Semmelweis University, 1085 Budapest, Hungary; 4ELKH-UD Vascular Pathophysiology Research Group 11003, University of Debrecen, 4032 Debrecen, Hungary

**Keywords:** stromal cell-derived factor-1, familial hypercholesterolemia, oxidized low-density lipoprotein, very low-density lipoprotein, large low-density lipoprotein, vascular repair

## Abstract

Stromal cell-derived factor-1 (SDF-1) is a chemokine that exerts multifaceted roles in atherosclerosis. However, its association with hyperlipidemia is contradictory. To date, serum SDF-1 and its correlations with lipid fractions and subfractions in heterozygous familial hypercholesterolemia (HeFH) have not been investigated. Eighty-one untreated patients with HeFH and 32 healthy control subjects were enrolled in the study. Serum SDF-1, oxidized LDL (oxLDL) and myeloperoxidase (MPO) were determined by ELISA. Lipoprotein subfractions were detected by Lipoprint. We diagnosed FH using the Dutch Lipid Clinic Network criteria. Significantly lower serum SDF-1 was found in HeFH patients compared to healthy controls. Significant negative correlations were detected between serum total cholesterol, triglycerides, LDL-cholesterol (LDL-C), apolipoprotein B100 (ApoB100) and SDF-1. Furthermore, serum SDF-1 negatively correlated with VLDL and IDL, as well as large LDL and large and intermediate HDL subfractions, while there was a positive correlation between mean LDL-size, small HDL and SDF-1. SDF-1 negatively correlated with oxLDL and MPO. A backward stepwise multiple regression analysis showed that the best predictors of serum SDF-1 were VLDL and oxLDL. The strong correlation of SDF-1 with lipid fractions and subfractions highlights the potential common pathways of SDF-1 and lipoprotein metabolism, which supports the role of SDF-1 in atherogenesis.

## 1. Introduction

Stromal cell-derived factor-1 (SDF-1), also known as chemokine C-X-C ligand 12 (CXCL12), is an important chemokine that is detectable in primary atherosclerotic plaques and exerts multiple roles in atherosclerosis through its classical C-X-C motif chemokine receptor 4 (CXCR4) receptor and alternative C-X-C motif chemokine receptor 7 (CXCR7) [1]. Recently, its participation in the atherosclerotic process has been extensively studied, however, its role seems to be diverse and has still not been fully clarified. Indeed, the function of the SDF-1-CXCR4 axis in native and accelerated, injury-driven atherosclerosis is ambivalent and highly context- and cell-dependent and may, therefore, confer protective as well as aggravating effects on the lesions’ formation [2,3]. Former studies have shown that SDF-1 may be involved in cellular adhesion by mediating the action of intercellular adhesion molecule-1 (ICAM-1) as it cooperates with ICAM-1 to capture progenitor cells from blood and induce the transendothelial migration of monocytes, suggesting a critical role of SDF-1 in atherogenesis [4]. However, data reveal that CXCR4 can act atheroprotectively in vascular cells by sustaining endothelial integrity and promoting the contractile phenotype of vascular smooth muscle cells. This conclusion is supported by the increase in atherosclerosis in both the arterial endothelial cell- and smooth muscle-specific deletion of CXCR4 in mice [5].

Some former studies suggested an association between serum SDF-1 and hyperlipidemia, however, the exact nature of the relationship seems to be uncertain. Serum SDF-1 was found to be elevated in male patients with mild hyperlipidemia but decreased in severe hyperlipidemia [6]. It was reported that hypercholesterolemia disturbs the bone marrow SDF-1/CXCR4 axis, generating a proatherogenic state in mice [7]. A previous study demonstrated low-density lipoprotein (LDL)-induced SDF-1 expression in endothelial cells and increased monocyte adhesion to endothelial cells in a dose-dependent manner [8]. SDF-1 overexpression in *ApoE*−/− mice was associated with increased macrophage infiltration, reduced reverse cholesterol transport, decreased plasma high-density lipoprotein-cholesterol (HDL-C) and the enlargement of atherosclerotic lesions. Moreover, the SDF-1/CXCR4 interaction inhibits the ABCA1-dependent cholesterol efflux from macrophages to apolipoprotein A1 (ApoA1) thus aggravating the atherosclerosis [9].

Familial hypercholesterolemia (FH) is a common autosomal disorder caused by genetic variants affecting the removal of LDL from plasma, resulting in early atherosclerotic cardiovascular disease. The lifetime cardiovascular risk of individuals with untreated heterozygous FH (HeFH) was estimated to be 3.9-fold greater than that of non-FH subjects [10]. This risk not only depends on the type of molecular defect of the canonical genes and consequent LDL-cholesterol (LDL-C) concentrations, but also on the presence of other traditional and non-traditional/novel risk biomarkers [11,12]. Since HeFH is characterized by severe hypercholesterolemia and enhanced atherogenesis, HeFH can be a suitable target population for examining serum SDF-1. To date, serum SDF-1 and its correlations with lipid, oxidative and inflammatory parameters, as well as lipoprotein subfractions, have not been conclusively studied. Therefore, we aimed to assess soluble ICAM-1 (sICAM-1), soluble vascular cell adhesion molecule-1 (sVCAM-1), the soluble form of CD40 ligand (sCD40L), oxidized LDL (oxLDL), tumor necrosis factor-alpha (TNFα), high sensitivity C-reactive protein (hsCRP) and myeloperoxidase (MPO) and LDL and HDL subfractions, as well as human paraoxonase-1’s (PON1) paraoxonase and arylesterase activities in a large, untreated HeFH population. Since some previous studies showed that the administration of statins may alter SDF-1 [13], we did not enroll patients already receiving statin treatment. A healthy normolipemic control population was also included.

We hypothesized that serum SDF-1 concentration is altered in untreated patients with HeFH and circulating SDF-1 correlates with serum lipoproteins, as well as inflammatory and oxidative parameters.

## 2. Results

Significantly higher total cholesterol, LDL-C, triglycerides, apoB100 and lipoprotein(a) (Lp(a)) were found in HeFH patients compared to control subjects, while circulating HDL-C and ApoA1 did not differ significantly. The mean LDL size was found to be significantly lower in HeFH. Significantly higher arylesterase activity of PON1 was found in HeFH compared to controls, but there were no differences in PON1’s paraoxonase and salt-stimulated paraoxonase activities between HeFH and controls. Circulating glucose, oxLDL, MPO and sICAM-1 were significantly higher, while TNFα was significantly lower in HeFH patients than in control subjects. There were no significant differences in body mass index (BMI), smoking status, hsCRP, sVCAM-1 and sCD40L between patients and controls (Table 1).

SDF-1 was significantly lower in HeFH compared to controls (71.3 ± 39.7 vs. 150.6 ± 55.4 pg/mL, respectively; *p* < 0.001) (Figure 1a and Table 1). Additionally, females had tendentiously higher serum SDF-1 than males (Figure 1b). We could not find significant differences in serum SDF-1 between females and males in HeFH patients (78.2 ± 39.9 vs. 59.1 ± 36.9 pg/mL, *p* = 0.07) and in controls (166.5 ± 51.3 vs. 129.8 ± 66.6 pg/mL, *p* = 0.14). There was no significant difference between the serum SDF-1 of smokers and non-smokers (97.0 ± 55.9 vs. 94.4 ± 68.7 pg/mL, *p* = 0.87).

The amount of LDL subfractions shifted towards larger LDL subfractions (Figure 2a), while we observed a shift towards smaller HDL subfractions (Figure 2b) in HeFH patients compared to the control population. The absolute amounts of VLDL and IDL subfractions were significantly higher in HeFH patients compared to control individuals. The concentrations of large- and small-density LDL subfractions were both significantly higher in HeFH compared to controls (Figure 2a). Furthermore, lower large and intermediate HDL subfractions, in contrast to a higher concentration of small HDL, was found in HeFH patients compared to control subjects (Figure 2b).

Significant negative correlations were detected between TC, triglycerides, LDL-C and ApoB100 in the whole study population (Figure 3, Table 2).

However, we could not find a significant correlation between HDL-C, ApoA1 and SDF-1. Furthermore, we found significant negative correlations between the amount of very low-density lipoprotein (VLDL), intermediate-density lipoprotein (IDL) and large LDL subfractions, while mean LDL size correlated positively with SDF-1 in the whole study population (Figure 4, Table 2).

SDF-1 positively correlated with large and intermediate HDL subfractions, while there was a negative correlation between SDF-1 and small HDL (Figure 5, Table 2).

Furthermore, SDF-1 negatively correlated with oxLDL (r = −0.51; *p* = 0.01) and the logarithm of serum MPO activity (lgMPO) (r = −0.33; *p* < 0.001) (Table 2). However, there were no significant correlations between other oxidative and inflammatory markers including either hsCRP, TNFα, sICAM-1, sVCAM-1, sCD40L or PON1’s aryesterase and paraoxonase activities.

Multiple regression analysis using a backward stepwise method showed that serum SDF-1’s best predictors turned out to be VLDL and oxLDL in both statistical models. The first model (Model 1) contained the TC, logarithm of serum triglyceride (lgTG), LDL-C, ApoB100, concentration of VLDL, IDL, large LDL, mean LDL size, large HDL, intermediate HDL, small HDL, oxLDL and lgMPO. Whereas in Model 2, the variables were selected on the basis of the biological traits of the items and contained the TC, logarithm of serum triglyceride (lgTG), ApoB100, concentration of VLDL, mean LDL size, large HDL, small HDL, oxLDL and lgMPO (Table 2).

Correlations in HeFH patients and controls were also calculated separately. We found similar tendencies in HeFH patients and controls. Significant positive correlations were found between mean LDL size, large HDL and SDF-1 in HeFH patients, while SDF-1 showed significant negative correlations with triglycerides and VLDL in controls (Appendix A).

## 3. Discussion

FH is an inborn error of LDL metabolism caused by mutations in the genes encoding the LDL receptor (*LDLR*), ApoB100 (*APOB*) or proprotein convertase subtilisin/kexin type 9 (*PCSK9*) [14,15]. HeFH carriers of a single mutation are relatively common; the condition globally occurs in one of every 200–300 people worldwide as well as in our region [16,17].

According to a large body of clinical evidence, a markedly elevated LDL-C and the particles’ accumulation in the arterial wall have been considered to promote cardiovascular risk in subjects with HeFH. Previously, several predictive score systems were suggested for FH patients, such as the FH-Risk-Score [18] and the SAFEHEART registry [19]. These were based on classical risk factors including TC and LDL-C. While these risk stratifications could be integrated into the clinical practice of HeFH, future investigation is needed to determine if other elements of risk stratification could improve HeFH outcomes. Therefore, in addition to LDL-C, evidence for other potential cardiovascular risk factors in patients with FH is also emerging [12]. Several former observations in HeFH populations failed to find associations between LDL-C and atherosclerotic cardiovascular events and indicated that other lipoprotein fractions and inflammatory cytokines might be strong driving factors of cardiovascular risk in the HeFH population [20,21,22]. Therefore, the potential role of novel factors, including recently identified and characterized chemokines, should be studied in this special patient population.

SDF-1, also known as CXCL12, is involved in inflammatory responses and neuromodulation in the brain by acting on its receptors CXCR4 and CXCR7 [23,24]. After a cerebral ischemic stroke, SDF-1α and CXCR4 are upregulated in the ischemic penumbra. Meanwhile, remote ischemic postconditioning increases SDF-1 in the peripheral blood, with the increase in the protein’s production being a possible part of the ischemic adaptation [25].

Previous studies have reported that the homing or recruitment of circulating endothelial progenitor cells (EPCs) to injury or ischemic sites by SDF-1 is an important process for executing their angiogenic and repair functions [26]. SDF-1 may be involved in regulating the mobilization, proliferation and adhesion capacity of EPCs through binding to CXCR4 and CXCR7 [27]. Of note, data indicate that the effects of SDF-1 on atherosclerosis rely on SDF-1’s production in arterial endothelial cells, identifying endothelial cell-derived SDF-1 as a crucial driver of atherosclerosis and an important contributor to the circulation of SDF-1. As this contribution only amounted to 25% of total plasma SDF-1, other cellular sources, e.g., adipocytes or hepatocytes, likely produce a substantial component of unknown functional relevance [3]. In our HeFH population, serum SDF-1 amounted to approximately half of the control population’s SDF-1. Therefore, the cause of decreased circulating SDF-1 in our HeFH patients could be impaired production by the endothelial cells accompanied by other mechanisms affecting SDF-1’s production. Although serum SDF-1 was moderately higher in females compared to males, the difference was not significant.

Previous data on the correlation between serum SDF-1 and lipid parameters are scant and contradictory. In a small previous study, serum SDF-1 was significantly higher in male patients with borderline dyslipidemia compared to control subjects. However, SDF-1 in male patients with clinically significant dyslipidemia was lower than in borderline dyslipidemia. Furthermore, there was a non-significant negative correlation between SDF-1 and the TC/HDL-C ratio. SDF-1 was positively associated with HDL-C in female patients only [6]. In another study, increasing quartiles of SDF-1 were associated with higher LDL-C and triglycerides, while HDL-C decreased across SDF-1 quartiles in participants of the Framingham Heart Study. Interestingly, the mean serum SDF-1 was 1894 pg/mL (range 742 pg/mL to 17,633 pg/mL), approximately five to 10 times higher than usually detected in other human studies [28]. In contrast, we found a strong negative correlation between SDF-1 and ApoB100-containing lipoproteins, and SDF-1 was predicted by VLDL. Recently, it has been shown that the hazard risk for having cardiovascular outcomes is greater for VLDL than LDL, since VLDL carries more cholesterol per particle than smaller LDL, and VLDL remnants are more easily trapped in the intima of the arterial wall, where they cause low-grade inflammation by direct and indirect mechanisms [29] and may suppress SDF-1’s expression as well. Although an elevated amount of triglyceride-rich particles is not characteristic of FH, the disorder is often associated with other conditions including overweight, which may lead to the elevation of triglyceride-rich particles such as VLDL and IDL, leading to elevated serum triglycerides. Formerly, data suggested that inflammatory cytokines (TNFα and IL-6) may have a suppressive effect on SDF-1 in male patients with hyperlipidemia [6]. Although we could not find correlations between inflammatory markers and SDF-1, higher hsCRP indicating enhanced systemic inflammation may diminish SDF-1’s production in HeFH.

To date, associations between lipoprotein subfractions and SDF-1 have not been studied. Significantly higher small dense LDL concentrations, lower mean LDL sizes, as well as lower large and intermediate HDL subfractions in contrast to higher small HDL was previously reported in HeFH patients compared to control subjects [22]. Based on our results, SDF-1 correlates positively with mean LDL size and large and intermediate HDL subfractions. However, the significance of these findings warrants further investigation.

OxLDL enhances coronary atherosclerosis by promoting cellularity, macrophage activation and differentiation into foam cells, stimulating smooth muscle cell proliferation, and decreasing endothelial nitric oxide (NO) production. MPO is one of the leading agents inducing oxidative stress, which is the basis for oxLDL-generation. OxLDL, or endogenous antibodies against oxLDL, have been found to relate to CVD, functioning as biomarkers [30]. However, its prognostic role in FH is not well established. The degree of LDL oxidation was not associated with the history of cardiovascular disease in adult male patients with HeFH, and there was no significant correlation between the plasma concentration of LDL-C and LDL’s degree of oxidation [31]. Moreover, no association was found between carotid intima-media thickness and oxidation parameters or circulating antibodies and oxLDL in a larger cohort of HeFH patients [32]. Another previous study, in which antibody titers were compared to oxLDL in patients with homozygous and heterozygous FH, demonstrated no significant relationship between the degree of atherosclerosis and the antibody titer [33]. However, the presence of scavenger receptor lectin-like oxLDL receptor-1 (LOX1)’s rs11053646 genotype enhanced the release of the soluble receptor, resulting in increased plaque instability and predicting coronary artery disease in adult patients with HeFH [34]. These incongruent results may demonstrate the fact that the interpretation of oxLDL can be challenging in clinical research. Indeed, oxLDL is a general term that covers heterogeneous oxidative changes to both LDL’s lipid moieties and ApoB. Of note, non-oxidizing modification of LDL, including desialylation, can also have an atherogenic effect, and thus should also be taken into consideration [35]. In line with the previous literature, serum oxLDL and MPO were significantly higher in our HeFH patients compared to controls [36]. Interestingly, we found negative correlations between oxLDL and both MPO and SDF-1 in our study population, which is a novel finding. Moreover, oxLDL was the strongest predictor of circulating SDF-1, suggesting a possible regulatory role of oxidative stress in SDF-1’s production.

It has been reported that atorvastatin increased SDF-1α’s expression in vivo under ischemia-reperfusion injury [13]. Moreover, SDF-1α’s upregulation by atorvastatin in rats with acute myocardial infarction via NO-production conferred anti-inflammatory and anti-apoptotic effects [37]. It has been reported that circulating microRNA miR-548j-5p contributes to the pathological process associated with angiogenesis by promoting migration and tube formation in EPCs, which are associated with the expression of eNOS and SDF-1. Therefore, upregulation of miR-548j-5p improved neovascularization, which implies that SDF-1 may be a potential therapeutic target for the treatment of peripheral artery disease (PAD) [38]. Furthermore, the NO-donor MPC-1011 stimulates angiogenesis and arteriogenesis and improves hindlimb ischemia via a cGMP-dependent pathway involving VEGF and SDF-1α [39]. Based on these data, there are some well-established and novel strategies for increasing circulating SDF-1 in HeFH, although its potential benefit in cardiovascular prevention needs further testing.

Some limitations of the study must be mentioned. The direct mechanisms of common regulatory pathways in SDF-1 and lipoprotein metabolism were not investigated. Furthermore, the Dutch Lipid Clinic Network diagnostic (DLCN) criteria system is one of the most widely used diagnostic algorithms for FH, which incorporates the LDL-C, clinical signs and family history of premature atherosclerotic cardiovascular disease, including coronary artery disease and PAD, to generate a score that leads to a classification of either “definite”, “probable” or “possible” FH. Moreover, the detection of a pathogenic DNA-mutation in any of the FH-related genes leads to a diagnosis of “definite FH”. However, some important limitations of the DLCN criteria must be noted. Clinical manifestations are infrequent, and family history is sometimes unavailable or unreliable. Moreover, DNA testing is often not readily available and, in some cases, not concordant with the FH phenotype [40]. Additionally, in young FH patients, the DLCN criteria might underestimate the probability of FH. The use of imaging modalities to identify and quantify the burden of atherosclerosis in the aorta, carotid arteries, coronary arteries and peripheral vasculature would improve the value of the study. However, the results underline the importance of studying the potential role of SDF-1 in the process of atherogenesis in HeFH.

## 4. Materials and Methods

### 4.1. Study Population

Eighty-one subjects (55 females and 26 males) with HeFH were enrolled in our study at the Lipid Outpatient Clinic of the Department of Internal Medicine, University of Debrecen, Hungary. All HeFH patients were heterozygous with a confirmed LDL receptor gene mutation or fulfilled the DLCN criteria for FH [41]. The patients were referred to our Lipid Outpatient Clinic by general practitioners, cardiologists and neurologists to verify the diagnosis of HeFH and initiate optimal therapy. We asked the patients to arrive after 12 h of fasting between 08:00–10:00 a.m. All HeFH patients were newly diagnosed without ongoing lipid-lowering drug treatment. The medical history of each patient was reviewed to identify previously diagnosed and treated acute myocardial infarction, stroke, carotid artery disease or PAD. Thirty-two gender- and age-matched healthy individuals were used as controls. In controls, the main inclusion criteria were: a normal body mass index; normal serum cholesterol, glucose and liver enzymes; currently not taking any medications; and no history of previous chronic or acute diseases during the past 3 months. Physical examination and electrocardiogram of controls did not show any abnormalities.

Exclusion criteria were type 1 or 2 diabetes mellitus, alcoholism, pregnancy, lactation, malignancy, known liver, endocrine and autoimmune diseases and chronic neurological and hematological disorders, which can be associated with peripheral neuropathy. All participants provided written, informed consent before enrollment. 

### 4.2. Sample Preparation and Routine Laboratory Analysis

Peripheral venous blood samples were taken after a 12 h overnight fast into Vacutainer tubes and sera were centrifuged after 30 min of rest at 3500 RPM for 15 min. Routine laboratory parameters including fasting glucose, hsCRP, triglycerides, total cholesterol, LDL-C, HDL-C and Lp(a) were determined using a Cobas c600 autoanalyzer (Roche Ltd., Mannheim, Germany) from the same vendor in the Department of Laboratory Medicine, Faculty of Medicine, University of Debrecen, Hungary. Tests were performed according to the recommendation of the manufacturer. The sera were kept frozen below −70 °C in 300 μL aliquots for subsequent laboratory assays.

### 4.3. Measurement of Inflammatory and Oxidative Markers

Serum SDF-1 was measured using a commercially available duoset enzyme-linked immunoassay (ELISA, Cat. No. DY350-05, R&D Systems Europe Ltd., Abington, UK) according to the recommendations of the manufacturer. The values were expressed as pg/mL. Intra-assay coefficients of variation ranged between 3.4–3.9% and inter-assay coefficients of variation were 8.2–13.4%. The assay range was 31.2–2000 pg/mL. Undiluted serum samples were used in this assay.

Circulating high-sensitivity TNF-α was measured from undiluted sera using a Quantikine TNF-α ELISA (R&D Systems Europe Ltd., Abington, UK). Intra-assay coefficients of variation ranged from 1.9 to 2.2% and inter-assay coefficients of variation ranged from 6.2 to 6.7%. Values were expressed as pg/mL.

Serum MPO concentration was measured by a sandwich ELISA technique (R&D Systems Europe Ltd., Abington, UK). Intra- and inter-assay coefficients of variation were 6.5–9.4% and values are expressed as ng/mL.

Serum oxLDL was assessed by ELISA (Cat. No. 10-1143-01, Mercodia AB, Uppsala, Sweden) in which two monoclonal antibodies were directed against separate antigenic determinants on the oxidized apolipoprotein B molecule. Intra- and inter-assay coefficients of variation were 5.5–7.3% and 4–6.2%, respectively. Values were presented as U/L and sera were used for a final dilution of 1/6561 according to the user’s manual.

Circulating sICAM-1, sVCAM-1 and sCD40L were measured with a sandwich ELISA (Cat. No. DCD540, DVC00 and DCDL40, respectively; R&D Systems Europe Ltd., Abington, UK) according to the manufacturer’s instructions. Intra-assay coefficients of variations were 3.7–5.2% (sICAM-1), 2.3–3.6% (sVCAM-1) and 4.5–5.4% (sCD40L), while the inter-assay coefficients of variation ranged between 4.4 and 6.7% (sICAM-1), 5.5 and 7.8% (sVCAM-1), 6.0 and 6.4% (sCD40L). All values of concentration were expressed as ng/mL.

### 4.4. Determination of PON1’s Enzyme Activities

Serum PON1’s paraoxonase and salt-stimulated paraoxonase activity were measured in Greiner microtiter plates using a kinetic, semiautomated method. Hydrolysis of paraoxon (O,O-diethyl-O-p-nitrophenyl phosphate, Sigma-Aldrich, Budapest, Hungary) as a substrate was followed at 405 nm at room temperature. Serum PON1’s arylesterase activity was determined using phenylacetate as a substrate (Sigma Aldrich, Budapest, Hungary), and the hydrolysis of the substrate was determined at 270 nm at room temperature, as previously described [42].

### 4.5. Determinations of Lipoprotein Subfractions

Circulating lipoprotein subfractions were detected using a commercially available polyacrylamide gel tube electrophoresis (Lipoprint System, Quantimetrix Corporation, Redondo Beach, CA, USA), as formerly described in detail [43,44]. Briefly, 25 µL frozen sera were pipetted into gel tubes with 200 and 300 µL loading gel containing lipophilic dye, followed by a half-hour photopolymerization. The gel tubes were then taken in an electrophoresis chamber with a constant of 3 mA/tube. Each electrophoretic cycle was loaded with a lipoprotein quality control provided by the manufacturer (Liposure Serum Lipoprotein Control, Quantimetrix Corp., Redondo Beach, CA, USA). After that, lipoprotein bands were scanned with an ArtixScan M1 digital scanner (Microtek International Inc., Hsinchu, Taiwan) and analyzed with the Lipoware Software Image SXM v.1.82 (2007) developed by Quantimetrix Corporation (Redondo Beach, CA, USA).

Up to seven LDL subfractions were detected between the VLDL and HDL peaks during the LDL-subfraction test (Cat. No. 48-7002). The proportion of large LDL (large LDL%) was defined as the summed percentages of LDL1 and LDL2, whereas the proportion of the small LDL (small-dense LDL%) was defined as the sum of LDL3–LDL7. Cholesterol concentrations of the LDL subfractions were determined by multiplying the relative area under the curve (AUC) of the subfractions by the total cholesterol concentration. The calculated total LDL-C was the sum of the cholesterol in midbands C through A (which are mainly comprised of IDL) plus the LDL subfractions (LDL1–LDL7). The calculated LDL-C correlated with the directly measured LDL-C (Lipoprint LDL: 130.8 ± 30.14 mg/dL vs. β-Quant LDL: 130.0 ± 30.42 mg/dL, r^2^ = 0.887), as described previously [45].

Ten HDL subfractions were determined during the HDL-subfraction test (Cat. No. 48-9002). Large (HDL1–3), intermediate (HDL4–7) and small (HDL8–10) HDL subfractions were detected between the VLDL/IDL/LDL and albumin bands. The cholesterol content of the HDL subfractions was also calculated using the Lipoware Software according to the relative AUC of the subfraction bands.

### 4.6. Statistical Analyses

Statistical analyses were performed using the Statistica 13.5.0.17 (TIBCO Software Inc., Tulsa, OK, USA), and graphs were made using the GraphPad Prism 6.01 (GraphPad Prism Software Inc., San Diego, CA, USA). We also calculated the statistical power with the SPH Analytics online calculator (SPH Analytics Ltd., Alpharetta, GA, USA) to validate the difference between circulating SDF-1 in HeFH patients (group 1) and in control subjects (group 2). The statistical power was above 0.8 (0.98). The difference between categorical variables was calculated with the chi-square test. The normality of the continuous variables was tested using the Kolmogorov–Smirnov test. The comparison between groups was analyzed with a Student’s unpaired *t*-test in case of normal distribution and the Mann–Whitney U-test was assessed in a non-parametric analyses. Data were expressed as mean ± SD or median (first and third quartiles). The relationship between continuous variables was examined with a Pearson’s test. Backward multiple regression analysis was performed to define which variable(s) is/are the best predictor(s) of serum SDF-1. The *p* ≤ 0.05 probability values were considered statistically significant.

## 5. Conclusions

This is the first clinical study demonstrating low SDF-1 in HeFH and its strong negative correlation with lipid fractions and subfractions in HeFH patients and controls. Our results in HeFH indicate that high oxLDL and other components of dyslipidemia can contribute to low circulating SDF-1, which can, in turn, disrupt angiogenic and vascular repair functions in this patient population, contributing to their high cardiovascular risk. It must be emphasized that these observational data cannot prove the causal link between SDF-1 and lipid homeostasis, but they definitely increase its possibility. The pathogenic and clinical significance of low circulating SDF-1 in HeFH needs further investigation.

Our results demonstrate the significance of a complex evaluation of novel cardiovascular risk factors in HeFH patients and highlight the importance of further studies to clarify the consequences of low SDF-1 concentrations in FH.

## Figures and Tables

**Figure 1 ijms-24-15308-f001:**
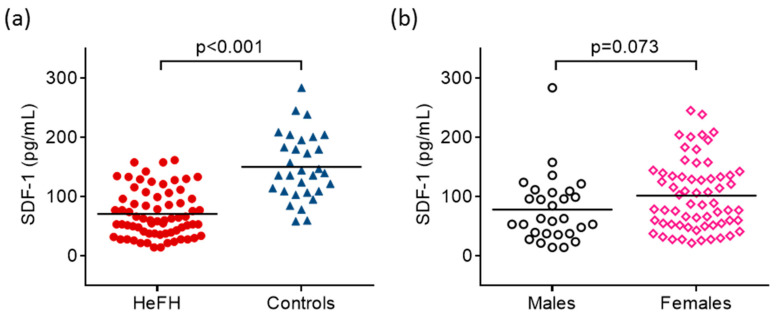
Serum concentrations of stromal cell-derived factor-1 (SDF-1) in patients with heterozygous familial hypercholesterolemia (HeFH) and controls (**a**). Serum concentrations of stromal cell-derived factor-1 (SDF-1) in males and females (**b**). SDF-1 concentrations were quantified from peripheral blood serum samples with an enzyme-linked immunosorbent assay.

**Figure 2 ijms-24-15308-f002:**
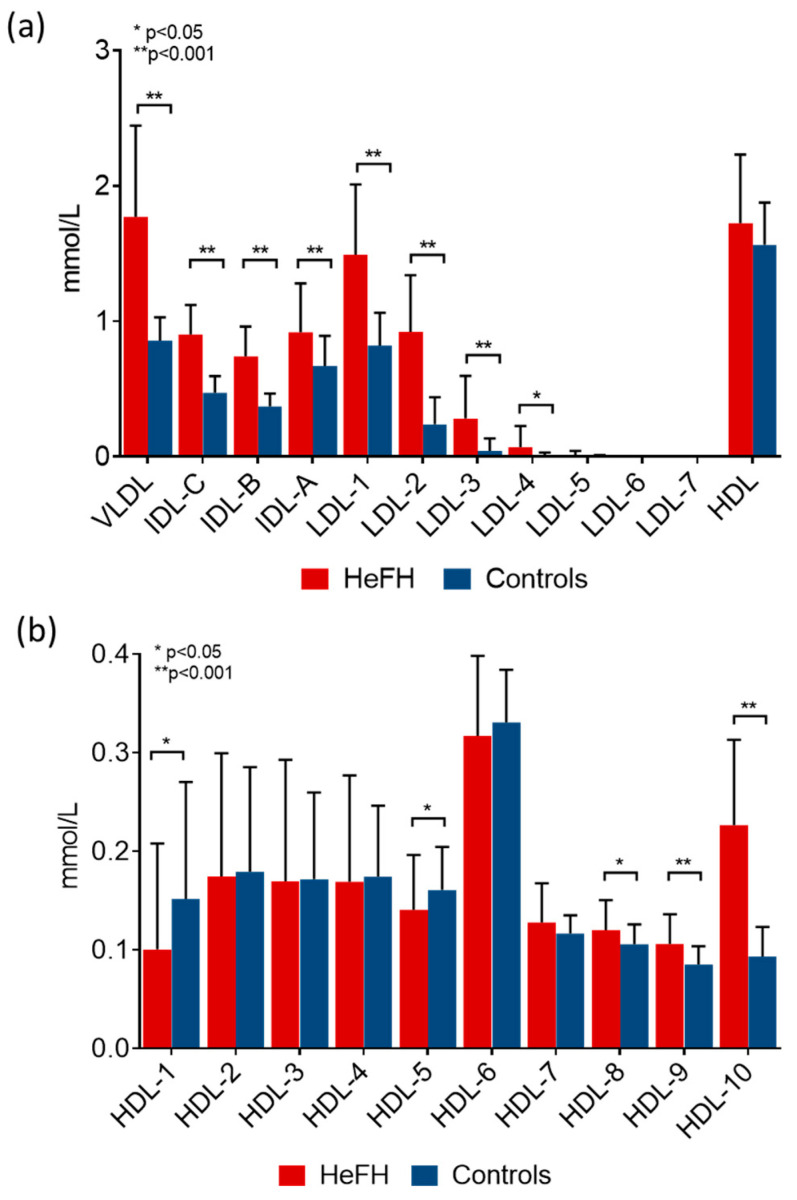
Amount of (**a**) lipoprotein fractions and (**b**) high-density lipoprotein (HDL) subfractions in patients with heterozygous familial hypercholesterolemia (HeFH) and controls (HeFH patients: red bar, n = 81; control subjects: blue bar, n = 32). Data are presented as mean ± standard deviation. Differences between the two study groups are analyzed using an unpaired *t*-test. * indicates *p* < 0.05; ** indicates *p* < 0.001 between HeFH and controls. LDL and HDL subfractions were analyzed by Lipoprint gel electrophoresis (Quantimetrix Corp. Redondo Beach, CA, USA) from peripheral blood serum samples.

**Figure 3 ijms-24-15308-f003:**
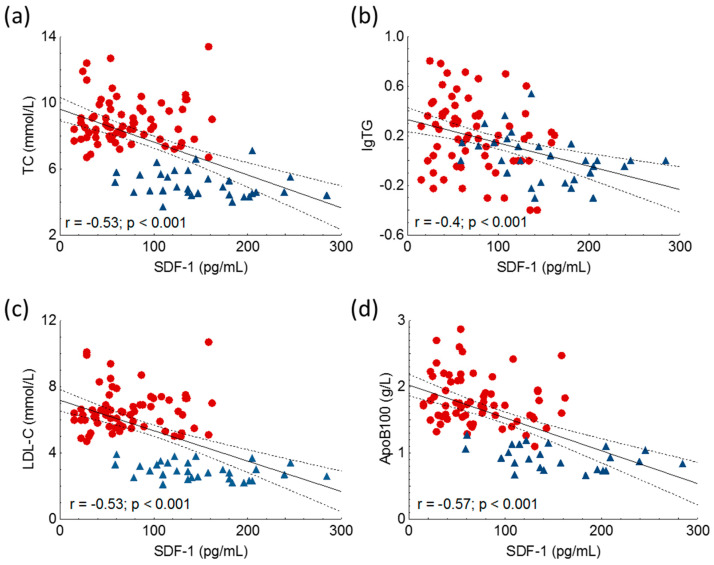
Correlations between serum level of total cholesterol (TC) (**a**), logarithm of triglycerides (lgTG) (**b**), low-density lipoprotein-cholesterol (LDL-C) (**c**), apolipoprotein B100 (ApoB100) (**d**) and stromal cell-derived factor-1 (SDF-1) in heterozygous familial hypercholesterolemia (HeFH) (red dots, n = 81) and controls (blue triangles, n = 32). SDF-1 concentrations were quantified from peripheral blood serum samples with an enzyme-linked immunosorbent assay.

**Figure 4 ijms-24-15308-f004:**
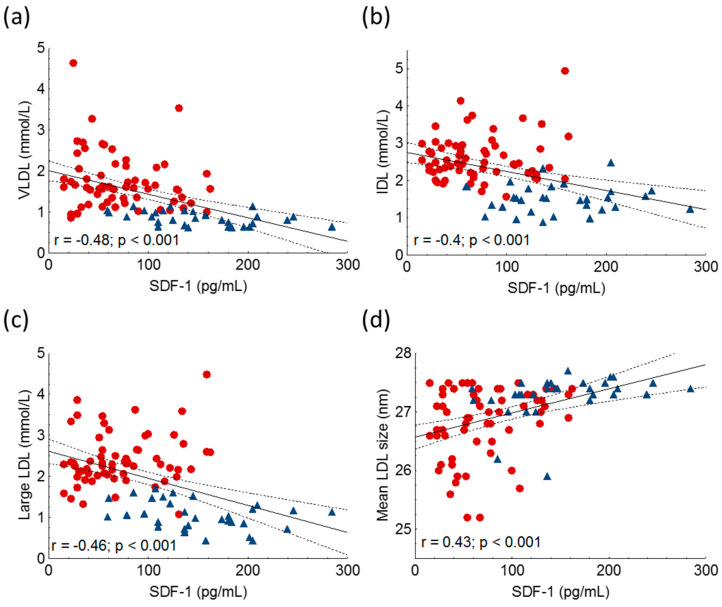
Correlations between serum very-low density lipoprotein (VLDL) (**a**), intermediate-density lipoprotein (IDL) (**b**), large low-density lipoprotein-cholesterol subfractions (large LDL) (**c**), mean LDL size (**d**) and stromal cell-derived factor-1 (SDF-1) in heterozygous familial hypercholesterolemic (HeFH) patients (red dots, n = 81) and controls (blue triangles, n = 32). SDF-1 concentrations were quantified from peripheral blood serum samples with an enzyme-linked immunosorbent assay. LDL subfractions and mean LDL size were detected by Lipoprint gel electrophoresis.

**Figure 5 ijms-24-15308-f005:**
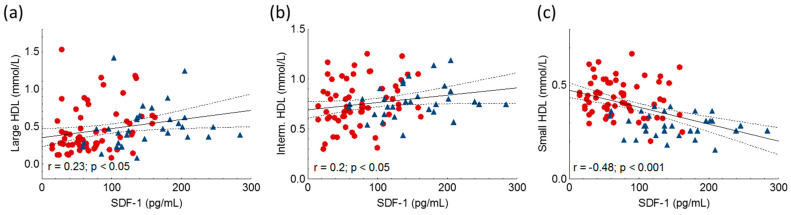
Correlations between large HDL (**a**), intermediate HDL (Interm. HDL) (**b**), small HDL (**c**) and stromal cell-derived factor-1 (SDF-1) in heterozygous familial hypercholesterolemic (HeFH) patients (red dots, n = 81) and controls (blue triangles, n = 32). SDF-1 concentrations were quantified from peripheral blood serum samples with an enzyme-linked immunosorbent assay. HDL subfractions were detected by Lipoprint gel electrophoresis.

**Table 1 ijms-24-15308-t001:** Anthropometric and laboratory parameters of study individuals. Values are presented as mean ± standard deviation or median (lower quartile-upper quartile).

	HeFH Patients	Controls	*p*-Values
Number of subjects	81	32	
Male/female	26/55	5/27	ns.
Age (yrs)	53.22 ± 14.5	41.8 ± 6.0	*p* < 0.001
Body mass index (kg/m^2^)	25.85 ± 3.67	24.48 ± 2.49	ns.
Fasting glucose (mmol/L)	5.03 ± 0.6	4.8 ± 0.5	*p* < 0.05
Smokers (n; %)	16 (19.75)	5 (16.13)	ns.
Lipid parameters			
Cholesterol (mmol/L)	8.87 ± 1.47	5.07 ± 0.78	*p* < 0.001
HDL-C (mmol/L)	1.62 ± 0.48	1.56 ± 0.46	ns.
LDL-C (mmol/L)	6.48 ± 1.28	2.93 ± 0.52	*p* < 0.001
Triglycerides (mmol/L)	1.6 (1.0–2.4)	1.0 (0.75–1.39)	*p* < 0.001
ApoB100 (g/L)	1.78 ± 0.38	0.94 ± 0.18	*p* < 0.001
ApoA1 (g/L)	1.71 ± 0.28	1.68 ± 0.31	ns.
Lp(a) (mg/L)	179 (75–857)	90 (30–214)	*p* < 0.05
Mean LDL size (nm)	26.78 ± 0.58	27.26 ± 0.37	<0.05
Inflammatory and oxidative markers			
hsCRP (mg/L)	1.84 (0.70–2.90)	1.55 (0.6–2.95)	ns.
PON1’s paraoxonase activity (U/L)	107.02 (43.61–166.5)	83.0 (47.9–167.4)	ns.
PON1’s salt-stimulated paraoxonase activity (U/L)	183.5 (103.2–322.6)	169.4 (97.3–297.4)	ns.
PON1’s arylesterase activity (U/L)	143.2 ± 25.12	135.4 ± 36.8	*p* < 0.01
MPO (ng/mL)	297.7 (158.15–456.5)	135.7 (99.4–195.1)	*p* < 0.001
oxLDL (U/L)	187.98 ± 71.04	41.1 ± 9.57	*p* < 0.001
sICAM-1 (ng/mL)	270.66 ± 69.9	210.8 ± 32.2	*p* < 0.001
sVCAM-1 (ng/mL)	573.9 ± 140.45	467.7 ± 106.3	ns.
sCD40L (ng/mL)	10.02 ± 4.3	8.22 ± 3.44	ns.
TNFα (pg/mL)	0.47 ± 0.17	1.66 ± 0.91	*p* < 0.001
SDF-1 (pg/mL)	71.3 ± 39.7	150.6 ± 55.4	*p* < 0.001

Abbreviations: ApoA1—apolipoprotein A1; ApoB100—apolipoprotein B100; HeFH—heterozygous familial hypercholesterolemia; HDL-C—high-density lipoprotein cholesterol; hsCRP—high sensitive C-reactive protein; LDL-C—low-density lipoprotein cholesterol; Lp(a) —lipoprotein (a); MPO—myeloperoxidase; ns.—non-significant; oxLDL—oxidized LDL; PON1—paraoxonase-1; sCD40L—soluble CD40 ligand; SDF-1—stromal cell-derived factor-1; sICAM-1—soluble intercellular adhesion molecule-1; sVCAM-1—soluble vascular adhesion molecule-1; TNFα—tumor necrosis factor alpha.

**Table 2 ijms-24-15308-t002:** Results of univariate and multivariate analyses between characteristics of all enrolled participants and serum stromal cell-derived factor-1 (SDF-1).

	Univariate Analysis		Multivariate Analysis Model 1		Multivariate Analysis Model 2	
	r	*p*-Value	β *	*p*-Value	β *	*p*-Value
Age (yrs)	−0.18	0.082				
Gender (m/f)	0.19	0.073				
Body mass index (kg/m^2^)	−0.21	0.061				
Fasting glucose (mmol/L)	−0.13	0.189				
Cholesterol (mmol/L)	−0.53	<0.001	0.123	0.548	0.126	0.485
lg Triglycerides (mmol/L)	−0.40	<0.001	0.161	0.355	0.105	0.503
LDL-C (mmol/L)	−0.53	<0.001	−0.01	0.993		
ApoB100 (g/L)	−0.57	<0.001	−0.11	0.770	−0.21	0.475
VLDL (mmol/L)	−0.48	<0.001	−0.49	<0.001	−0.48	<0.001
IDL (mmol/L)	−0.40	<0.001	−0.04	0.888		
Large LDL (mmol/L)	−0.46	<0.001	−0.001	0.995		
Mean LDL size (mmol/L)	0.43	<0.001	−0.08	0.675	−0.01	0.945
HDL-C (mmol/L)	0.07	0.496				
ApoA1	0.01	0.891				
Large HDL (mmol/L)	0.23	<0.05	−0.16	0.453	−0.03	0.873
Intermediate HDL (mmol/L)	0.20	<0.05	0.085	0.462		
Small HDL (mmol/L)	−0.48	<0.001	−0.22	0.063	−0.24	0.067
Oxidized LDL (U/L)	−0.51	0.01	−0.31	<0.001	−0.32	0.006
lg MPO	−0.33	<0.001	−0.13	0.279	−0.13	0.275

Abbreviations: ApoA1—apolipoprotein A1; ApoB100—apolipoprotein B100; HDL-C—high-density lipoprotein cholesterol; IDL—intermediate-density lipoprotein; LDL-C—low-density lipoprotein cholesterol; MPO—myeloperoxidase; oxLDL—oxidized LDL; VLDL—very-low density lipoprotein. Notes: The relationship between laboratory variables and SDF-1 was analyzed with Pearson’s univariate test. Backward stepwise multiple regression analysis was performed to define significant predictor(s) of serum SDF-1. Model 1 included all variables, which showed a correlation with SDF-1 in the univariate analysis; whereas in Model 2, the variables were selected on the basis of the biological traits of the items. β * means the standardized beta.

## Data Availability

All data generated or analyzed during this study are included in this published article. All data generated or analyzed during the current study are available from the corresponding author upon reasonable request.

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
