# Peer review of "Decreased Serum Stromal Cell-Derived Factor-1 in Patients with Familial Hypercholesterolemia and Its Strong Correlation with Lipoprotein Subfractions"

_ijms, 2023, doi:10.3390/ijms242015308_

Round 1
Reviewer 1 Report
The study seemed to include some interesting findings. The improvement of the paper may be further considered.
1. Were the controls appropriate for the study design. If the traits of FH are assessed, the atherosclerotic factors such as glucose concentrations, body mass index levels and gender of controls should be matched with those of FH. If not, the conclusions can be different from the hypothesis.
2. It may be due to the suggestion 1 (the setting of super-healthy controls); the readers might have an odd impression to the result of the significant correlation between VLDL and SDF-1 in FH as the pathology of FH is not very much related to VLDL and triglycerides.
3. In the study design, age and gender ratio should be fully matched because the characteristics largely affect the atherosclerosis. In terms of gender, the gender-separated sub-analysis is also considered.
4. The lifestyles, such as smoking and exercise habits, should be adjusted as smoking in specific is a traditional atherosclerotic and oxidative factor.
5. We would rethink the specificity of the conclusion to FH. The results should be tested in type II hyperlipidemia (WHO classification) using the patient groups with non-FH but mild-to-moderate high concentrations of LDL-C. This may be also related to the consideration of the suggestion 1 (the setting of controls).
6. In the methods, the very small numbers of stroke and some overt atherosclerotic diseases were included in the study. Those should be excluded for definitive conclusions. The significance due to the small numbers may not be counted on in some cases.
7. In the methods, the withdraw periods of lipid-lowering drugs could be described as the recruitment criteria.
8. In the methods, the assay coefficients could be described in SDF-1 measurement. The SDF-1 is a center measure of the study.
9. In the methods, correlations were analyzed in a single group mixed of clearly different populations of healthy controls and FH. That may be a sense of discomfort for readers. Please compare respectively the correlations of the groups separated of controls and FH.
10. The results of the simple and multivariate analysis for correlations should be presented in Table.
11. In FH, are the LDL-C or oxLDL levels predictive to cardiovascular events in general? If so, the implication of correlation with SDF-1 can be more discussed using such information with literature.
12. In the Discussion, the limitation of Dutch criteria for diagnosis of FH could be described.
13. The mixed expression of apo/Apo was seen to be unified; that is, Raw 23; Apolipoprotein, Raw 24; ApoB100, Raw 63; apolipoprotein, Raw 86; apoB100, Raw 179…. Recheck the overall text.
14. The repeated expression of abbreviations was seen to be unified; that is, Raw 202 and 255; EPCs, Raw 116 and 258; PAD.
15. The abbreviation of FH was canceled (why?); Raw 401.
The use of abbreviations should be rechecked. There found to be an incorrect use of adverbs and conjunctions. Native check is a must-have.
Author Response
Dear Reviewer, thank you for your positive reply and helpful comments on our manuscript. We hereby answer your recommendations as follows:
- Were the controls appropriate for the study design. If the traits of FH are assessed, the atherosclerotic factors such as glucose concentrations, body mass index levels and gender of controls should be matched with those of FH. If not, the conclusions can be different from the hypothesis.
Response: Thank you for the valuable comment. According to the suggestion of the reviewer, we evaluated the data on the above mentioned atherosclerotic factors including glucose concentration and body mass index based on the source documentation. These data were added to Table 1 (Ln 97). Data on gender have been included in the originally submitted manuscript (Table 1).
Based on the documentation the ratio of smokers vs. non-smokers was 16 vs. 64 (19.75 %) in the HeFH population, and 5 vs. 26 (16.13 %) in the control population, without a statistically significant difference between the two study groups (p=0.43). Mean BMI was 25.85±3.67 kg/m2 in the HeFH group and 24.48±2.49 kg/m2 in controls, the difference was statistically not significant (p=0.09). Based on these results, our control population is gender- and BMI-matched. This information has been added to the “Results” section (Ln 94).
- It may be due to the suggestion 1 (the setting of super-healthy controls); the readers might have an odd impression to the result of the significant correlation between VLDL and SDF-1 in FH as the pathology of FH is not very much related to VLDL and triglycerides.
Response: The reviewer is right. In FH the classical lipid abnormalities are elevations of serum total and LDL-cholesterol. However, FH is commonly associated with other conditions including overweight, which may lead to the elevation of triglyceride-rich particles such as VLDL and IDL, leading to an elevated serum triglyceride. The prevalence of overweight and obesity is increasing worldwide, and is commonly seen in the Eastern part of Europe as well. Based on the results of a previous study conducted by our research group using data mining and machine learning, the ratio of obesity in Hungarian FH-patients (n=459) was 40% (Nemeth A et al, 2022) which underlines the significance of examining concomitant atherosclerotic risk factors that might also worsen the individual patient’s risk profile and highlights the complexity of FH-care. Indeed, these additive risk factors are summarized by our research group in a recently published review (Berta E et al, 2022). Average BMI of the HeFH group was mildly above the normal range (18.5-24.9 kg/m2) suggesting the additive role of overweight leading to triglyceride- and VLDL-elevation. Although mean BMI was in the normal range in controls, there were individual participants with elevated BMI-values. To clarify the appearance of elevated VLDL and triglyceride in HeFH we amended the “Discussion” section (Ln 271-274).
Németh Á, Daróczy B, Juhász L, Fülöp P, Harangi M, Paragh G. Assessment of Associations Between Serum Lipoprotein (a) Levels and Atherosclerotic Vascular Diseases in Hungarian Patients With Familial Hypercholesterolemia Using Data Mining and Machine Learning. Front Genet. 2022 Feb 9;13:849197. doi: 10.3389/fgene.2022.849197. eCollection 2022.
Berta E, Zsíros N, Bodor M, Balogh I, Lőrincz H, Paragh G, Harangi M. Clinical Aspects of Genetic and Non-Genetic Cardiovascular Risk Factors in Familial Hypercholesterolemia. Genes (Basel). 2022 Jun 27;13(7):1158. doi: 10.3390/genes13071158.
- In the study design, age and gender ratio should be fully matched because the characteristics largely affect the atherosclerosis. In terms of gender, the gender-separated sub-analysis is also considered.
Response: Thank you for the comment. In clinical studies the enrollment of an optimal control population is probably the hardest part of the work. Our study population is matched for gender, BMI and ratio of smokers, but younger than the HeFH patient group. The main cause of it is that we could not find healthy subjects of this age, which might reflect the disappointing heath condition of the average Hungarian population, which might be responsible for the disillusive statistical data on cardiovascular morbidity and mortality in our region. It must be noted that there was no significant correlation between age and SDF-1 (Table 2). Therefore, the difference in age between the two study groups may not influence our conclusions.
In terms of gender, we made the gender-separated sub-analysis according to the reviewer’s suggestion. We could not find significant differences in serum SDF-1 between females and males in the whole study population (102.0±58.3 vs.78.6±55.8 pg/mL, p=0.07), in HeFH patients (78.2±39.9 vs. 59.1±36.9 pg/mL, p=0.07) or in controls (166.5±51.3 vs. 129.8±66.6 pg/mL, p=0.14), although SDF-1 was tendentiously higher in females. Results found in the whole study population were added to Figure 1 (Figure 1b, Ln 114). Results of the subgroups are added to the “Results” section (Ln 108-113) and commented in the Discussion (Ln 244-245).
- The lifestyles, such as smoking and exercise habits, should be adjusted as smoking in specific is a traditional atherosclerotic and oxidative factor.
Response: Thank you for the valuable comment. The ratio of smokers was calculated, and did not differ significantly in HeFH patients and controls (see above, Table 1). Moreover, there was no significant difference between serum SDF-1 of smokers and non-smokers (97.0±55.9 vs. 94.4±68.7 pg/mL, p=0.87). These data were added to the “Results” section (Ln 111-113). Because of the low number of smokers, data of the subgroups are not included. Unfortunately, creditable data on exercise habits are not available. Regarding oxidative factors, oxLDL and MPO were measured. In line with the literature, we found significantly higher oxLDL and MPO in HeFH patients compared to controls (Table 1), and the correlations between oxLDL and SDF-1, and between lgMPO and SDF-1 were statistically significant (Ln 184-188). Therefore, these factors are included in the multiple regression analysis, and oxLDL was found to be a significant predictor of SDF-1 level oxLDL (β=-0.31; p<0.001). (Table 2, Ln 190).
- We would rethink the specificity of the conclusion to FH. The results should be tested in type II hyperlipidemia (WHO classification) using the patient groups with non-FH but mild-to-moderate high concentrations of LDL-C. This may be also related to the consideration of the suggestion 1 (the setting of controls).
Response: Thank you for this forward-looking comment. Measurement of SDF-1 in non-FH patients with mild-to-moderate high concentrations of LDL-C and evaluation of its correlations with lipid parameters could definitely strengthen our finding. Therefore, we plan to continue the work in patients with various types of hyperlipidemia. However, these clinical studies need ethical approval, which is currently not available for other patient populations. Application for ethical approval, enrollment of a new patient population, laboratory analysis of the samples and statistical evaluation of the data are time-consuming and incurs costs. Therefore, these further researches are beyond the scope of the current work. However, thesis type of hyperlipidemic patients has to be considered for future clinical studies.
- In the ”Methods” section, the very small number of strokes and some overt atherosclerotic diseases were included in the study. Those should be excluded for definitive conclusions. The significance due to the small numbers may not be counted on in some cases.
Response: The reviewer is right. Data on stroke patients are deleted because of the small number of patients in this subgroup (Ln 121-124, 228-232).
- In the methods, the withdraw periods of lipid-lowering drugs could be described as the recruitment criteria.
Response: Thank you for the comment. Indeed, our patients were newly diagnosed without ongoing lipid-lowering drug treatment. Therefore, there were no withdraw periods (Ln 348-350)
- In the methods, the assay coefficients could be described in SDF-1 measurement. The SDF-1 is a center measure of the study.
Response: Thank you for the suggestion. Missing assay coefficients were added to the “Methods” section (Ln 388-389)
- In the methods, correlations were analyzed in a single group mixed of clearly different populations of healthy controls and FH. That may be a sense of discomfort for readers. Please compare respectively the correlations of the groups separated of controls and FH.
Response: Thank you for the valuable comment. Results of univariate analysis for correlations in HeFH and controls are presented separately in Supplementary Table S1.
- The results of the simple and multivariate analysis for correlations should be presented in Table.
Response: Thank you for the suggestion. Results of the simple and multivariate analysis for correlations are presented in a new table (Table 2).
- In FH, are the LDL-C or oxLDL levels predictive to cardiovascular events in general? If so, the implication of correlation with SDF-1 can be more discussed using such information with literature.
Response: Thank you for the comment. Since the topic is highly important and extensively studied, the predictive role of LDL-C and oxLDL as well as other modified LDL-forms are discussed in more detail in the “Discussion” section (Ln 285-312).
- In the Discussion, the limitation of Dutch criteria for diagnosis of FH could be described.
Response: Thank you for the comment, according to the reviewer’s suggestion, the main limitations of the Dutch criteria for the diagnosis of FH were summarized in the “Discussion” secion (Ln 327-338).
- The mixed expression of apo/Apo was seen to be unified; that is, Raw 23; Apolipoprotein, Raw 24; ApoB100, Raw 63; apolipoprotein, Raw 86; apoB100, Raw 179…. Recheck the overall text.
Response: Sorry for the inaccuracies. Abbreviations were rechecked and corrected throughout the manuscript.
- The repeated expression of abbreviations was seen to be unified; that is, Raw 202 and 255; EPCs, Raw 116 and 258; PAD.
Response: These abbreviations were also checked, unified and corrected according to the reviewer’s suggestion.
- The abbreviation of FH was canceled (why?); Raw 401.
Response: This sentence is the final conclusion of the work, so we felt that familial hypercholesterolemia could sound better than an abbreviation (FH). However, we corrected the sentence according to the reviewer’s suggestion (Ln 503).
Comments on the Quality of English Language
The use of abbreviations should be rechecked. There found to be an incorrect use of adverbs and conjunctions. Native check is a must-have.
Response: Abbreviations were rechecked and corrected throughout the manuscript. The text was checked by a native speaker.
Again, we are very thankful for your valuable and thorough review.
Reviewer 2 Report
This cross-sectional clinical study provides a generally coherent dataset.
1. Observational data only generate a hypothesis and result in correlations that may reflect coincidence, cause, consequence, or confounding. There is no proof that SDF-1 is on the causal pathway between lipoproteins and atherosclerosis.
2. An important pitfall of the correlation coefficient is that it is influenced by the range of observations. The correlations in Figure 3 are heavily dependent on the combination of the two groups. What are the correlation coefficients when the familial hypercholesterolemia patients and the controls are analyzed separately?
3. Same remark for Figure 4 and Figure 5. The correlation coefficients tend to be high because two completely distinct groups are combined.
4. This is a cross-sectional study. How were controls selected?
Author Response
Dear Reviewer, thank you for your helpful comments and suggestions on our manuscript. We hereby answer your recommendations as follows:
- Observational data only generate a hypothesis and result in correlations that may reflect coincidence, cause, consequence, or confounding. There is no proof that SDF-1 is on the causal pathway between lipoproteins and atherosclerosis.
Response:
Thank you for the comment. The reviewer is right; our observational data only generate hypothesis. Still, this is the first clinical study reporting low SDF-1 in HeFH and its strong negative correlation with lipid fractions and subfractions in HeFH patients and controls. Although they may reflect coincidence, cause, consequence, or confounding, the result are thought-provoking and the result of the multivariate analysis may indicate a connection between lipid metabolism and SDF-1, directly or indirectly. Indeed, we would like to highlight a possible link to induce further studies clarifying the exact mechanisms that may explain these observations. Therefore, our conclusions were refined to clarify the limitations of the observational data (Ln 496-498).
- An important pitfall of the correlation coefficient is that it is influenced by the range of observations. The correlations in Figure 3 are heavily dependent on the combination of the two groups. What are the correlation coefficients when the familial hypercholesterolemia patients and the controls are analyzed separately?
Response: Thank you for the suggestion. According to the suggestion, correlation coefficients were calculated in both study populations and the result are added in Supplementary table 1. Although we found similar tendencies in HeFH patients and controls, these correlations were mostly not significant. Theoretically, in larger patient and control populations these correlations would be significant as well. Data are added to the Results (Ln 199-203).
- Same remark for Figure 4 and Figure 5. The correlation coefficients tend to be high because two completely distinct groups are combined.
Response: Again, thank you for this comment. According to the suggestion, correlations in Figure 4 and 5 were calculated as well. Although we found similar tendencies in HeFH patients and controls, these correlations were not significant. Theoretically, in larger patient and control populations these correlations would be significant as well. Data are shown in Supplementary table 1 and added to the Results (Ln 199-203).
- This is a cross-sectional study. How were controls selected?
Response: Thank you for the valuable comment. In clinical studies the enrollment of an optimal control population is probably the hardest part of the work. Control were enrolled at the General Internal Medicine Outpatient Clinic of the Department of Internal Medicine, University of Debrecen. These subject were investigated because of common symptoms such as headache, palpitation, chest or leg pain, but based on the finding of the physical examination, ECG and laboratory test we could not identify any abnormalities. In our controls, the main inclusion criteria were normal body mass index; normal serum cholesterol, glucose and liver enzymes; currently not taking any medications; and no history of previous chronic or acute diseases during the past 3 months. Physical examination and electrocardiogram of controls did not show any abnormalities (Ln 355-359).
Again, we are very thankful for your valuable and thorough review.
Round 2
Reviewer 1 Report
The paper was improved, but there remained to be crucial parts reconsidered.
1. The results of multivariate analysis in Table 2 might have some multicollinearity (due to the comparison of plus/minus of r and beta in each variable item, the comparison of beta-values and p-values among each variable item). Although statistical way is important, the variable items used in the multivariate analysis can be often selected on the basis of biological traits of variable items.
2. Supplementary Table S1 was added. Could you please compare statistically the correlation in each variable item between HeFH and control groups?
Author Response
Dear Reviewer,
Thank you for your positive reply and helpful comments on our revised manuscript. We hereby answer your recommendations as follows:
- The results of multivariate analysis in Table 2 might have some multicollinearity (due to the comparison of plus/minus of r and beta in each variable item, the comparison of beta-values and p-values among each variable item). Although statistical way is important, the variable items used in the multivariate analysis can be often selected on the basis of biological traits of variable items.
Answer: Thank you for your comment. Basically, during the backward stepwise multivariable analysis, we included all variable what showed significant correlations with SDF-1 by univariate analysis. However, according to your suggestion, we selected among these variables on the basis of biological traits and the results of the second multivariable analysis were included into Table 2 as Model 2. In line with the first analysis, VLDL (β=-0.48; p<0.001) and oxLDL (β=-0.32; p=0.006) were significant predictors of SDF-1 (Table 2, Ln 180-188).
- Supplementary Table S1 was added. Could you please compare statistically the correlation in each variable item between HeFH and control groups?
Answer: Thank you for your forward-looking comment. When conducting correlation analyses by two independent groups (HeFH and controls), the comparison between the two correlations can be examined by z-score analysis. However, it must be noted, that z-score analysis is recommended when the correlations are conducted on the same variables by two different groups, and if both correlations are found to be statistically significant. In our case, there was no correlation that was statistically significant (p<0.05) in both study group in parallel. Therefore, we could not perform the above-mentioned analysis. Please find a description about z-score analysis below. Again, thank you for the valuable comment.
https://www.statisticssolutions.com/comparing-correlation-coefficients/
https://www.psychometrica.de/correlation.html
Once again, we are grateful for your valuable comments and suggestions, which have greatly enhanced the quality of our study.
Reviewer 2 Report
The authors have provided an ad rem answer to my comment and have changed the manuscript sufficiently.
Author Response
Dear Reviewer,
Once again, we are grateful for your valuable comments and suggestions, which have greatly enhanced the quality of our study.